# Virulence Diversity of *Puccinia striiformis* f. sp. *Tritici* in Common Wheat in Russian Regions in 2019–2021

Elena Gultyaeva [1,*], Ekaterina Shaydayuk [1] and Evsey Kosman [2]

1   All Russian Institute of Plant Protection, Shosse Podbelskogo 3, St. Petersburg 1986608, Russia
2   Institute for Cereal Crops Improvement, School of Plant Sciences and Food Security, George S. Wise Faculty of Life Sciences, Tel Aviv University, Tel Aviv 69978, Israel
*   Correspondence: egultyaeva@vizr.spb.ru

**Abstract:** Yellow (stripe) rust, caused by *Puccinia striiformis* f. sp. *tritici* (*Pst*), is a major disease of common wheat worldwide. Disease epidemics in Russia have been frequent and destructive, mostly in the North Caucasus. However, over the last 5 years, the significance of *Pst* has markedly increased in other Russian regions. Therefore, the *Pst* virulence diversity was investigated in *Triticum aestivum* in six geographically distant regions of the European (North Caucasus, North-West, Low Volga, Central Black Earth region, and Volga-Vyatka) and Asian (West Siberia) parts of Russia, with strongly different climates, environmental conditions, and growing wheat genotypes. Seventy-nine virulence pathotypes among 117 isolates were identified using the 12 Avocet *Yr* gene lines (*Yr1*, *Yr5*, *Yr6*, *Yr7*, *Yr8*, *Yr9*, *Yr10*, *Yr15*, *Yr17*, *Yr24*, *Yr27,* and *YrSp*) and eight supplemental wheat differentials (Heines VII, Vilmorin 23, Hybrid 46, Strubes Dickkopf, Carstens V, Suwon 92/Omar, Nord Desprez, and Heines Peko). Only four pathotypes occurred in two or more regions. High variability was detected within *Pst* populations from Dagestan, Central, North-West, and West Siberia that postulated to form an intrapopulation subdivision of each of them into several subgroups. Most regional virulence groups of pathotypes were closely related, except for several small subgroups of pathotypes from West Siberia, Dagestan, North-West, and Central European regions. All *Pst* isolates were avirulent in lines with *Yr5*, *Yr10*, *Yr15,* and *Yr24* genes. Virulence to *Yr17* was detected for several isolates of two pathotypes, one each from the North-West and Low Volga regions. Variation in virulence frequency was observed in other differential lines.

**Keywords:** population structure; race composition; *Triticum aestivum*; virulence; yellow rust; *Yr* genes

## 1. Introduction

Wheat *(Triticum aestivum)* is the main strategic agricultural crop [1]. Epidemics caused by plant pathogens can significantly affect yield and grain quality of wheat. Until recently, leaf rust was the most frequent, widespread, and harmful disease of wheat worldwide [2]. In the 2000s, the situation changed and, in many countries, outbreaks of wheat yellow rust (also known as stripe rust) has become the most hazardous [3]. The causative agent of yellow rust infects more than 20 species of cultivated and wild cereals, but it is more damaging for wheat, triticale, barley, and rye. The specialized form infecting wheat is referred to as *Puccinia striiformis* f. sp. *tritici* (*Pst*) [4,5]. *Pst* reproduces primarily clonally, although a sexual stage on *Berberis* spp. and *Mahonia* spp. has been recently discovered [4,6–9].

Yellow rust is considered to be a low-temperature disease and frequently occurs in temperate areas with cool, humid summers or in warm, high-altitude areas with cool nights [4,10]. Until the 2000s, the interregional impact of *Pst* was relatively limited compared with the more widely adaptable species, *P. triticina* and *P. graminis*, and wheat yellow rust was of only regional significance throughout the world. However, over the last 20 years, *Pst* has become one of the largest biotic limitations to wheat production and

threatens global food supply [11]. This is due to the emergence and rapid spread of new aggressive isolates—adapted to high temperatures, which have been able to produce more spores and induce disease symptoms more quickly [12]—as well as high mutation rate of the pathogen for virulence. Like other *Puccinia* spp., *Pst* has a high dispersal capacity, which contributes to the rapid spread of new races over long distances [13]. Yellow rust epidemics occur frequently in Western Europe, Central and East Asia, the Middle East, North and South Africa, North and South America, and Australia [3,4,10,13–18]. Yield losses in these widespread epidemics ranged from 10 to 70% [13,19] and were especially high when the epidemics were associated with exotic temperature-adapted aggressive strains, as these were rarely considered in breeding for disease resistance locally [18,20]. The highly aggressive race, *PstS1*, was detected as an exotic incursion in the USA in 2000 and in Australia in 2002, and race *PstS2* was first detected in Europe in 2000 [21,22]. Since 2011, invasive strains of the Warrior and Kranich races have largely replaced the pre-existing northwestern European populations [23,24]. These were later designated as *Pst*S7 and PstS8, respectively [21], and were detected across Europe and infected cultivars with durable adult plant resistance. Both races have been found to be distinct from the typical European isolates in that they produced an unusually high number of teliospores [23,24]. Currently, Warrior race, which occurs at a low frequency, is recorded in Europe, northern Africa, and South America. In the mid-2010s, additional aggressive *Pst* races have been determined (e.g., *PstS10* or Warrior(-) and *PstS4* Triticale aggressive), and they are now dominant in Europe [21,25,26].

Growing of resistant cultivars is the most economically effective method for yellow rust management. However, within a few years of deploying cultivars with race-specific resistance genes, new virulent races (pathotypes) of the rust pathogen emerge, able to infect previously resistant cultivars. Monitoring *Pst* races is of a high priority in all epidemic regions in the world because the rust urediniospores can be wind-transported over long distances [27]. *Pst* surveys have been conducted all over the world, with corresponding national programs established in Europe, North America, Australia, China, and other regions after severe epidemics [10,18,20]. Virulence of the yellow rust pathogen has been studied at the international scale by the Global Rust Reference Center (GRRC), Aarhus University, Denmark. However, Russian *Pst* populations were not included.

Until recently, yellow rust epidemics in Russia were frequent and destructive primarily only in the North Caucasus [28]. However, significance of *Pst* remarkably increased over the last 5 years. The disease has appeared regularly in other Russian regions (North-West, West Siberia, Volga, and Central Black Earth) [29–31]. Global climate changes contribute to alteration of the disease incidence and severity.

Virulence characterization of *Pst* populations provides information on effectiveness of resistance genes, but identification of new *Pst* races is needed for screening wheat germplasm for resistance. In Russia, only one *Pst* population from common wheat in the North Caucasus region (Krasnodar, Stavropol, and Rostov) has been studied [32]. The present research was designed to study *Pst* virulence pathotypes in geographically dispersed regions of Russia. The objectives were to (1) assess virulence variability and race composition of *Pst* isolates infecting common wheat in distant Russian locations and (2) compare the pathogen structure of regional populations.

## 2. Materials and Methods

### 2.1. Yellow Rust Sampling and Spore Multiplication

During the 2019–2021 cropping seasons, common wheat leaves with *Pst* uredinia were collected from disease nurseries, breeding plots, and commercial fields in six Russian agroecological regions, North Caucasus (Dagestan, Krasnodar, and Rostov, Russia), Low Volga (Saratov, Russia), North-West (St. Petersburg, Russia), Central Black Earth (Tambov, Russia), Volga-Vyatka (Kirov, Russia), and West Siberia (Novosibirsk and Krasnoyarsk, Russia) (Figure 1). In total, 95 *Pst* samples of air-dried infected leaves were obtained. One to ten leaves of a single cultivar from each plot/field were considered as one sample.

Sampling details of *Pst* isolates are given in Table 1. Leaves with *Pst* were initially stored at 4 °C with urediniospores from these samples revived and multiplied as soon as possible.

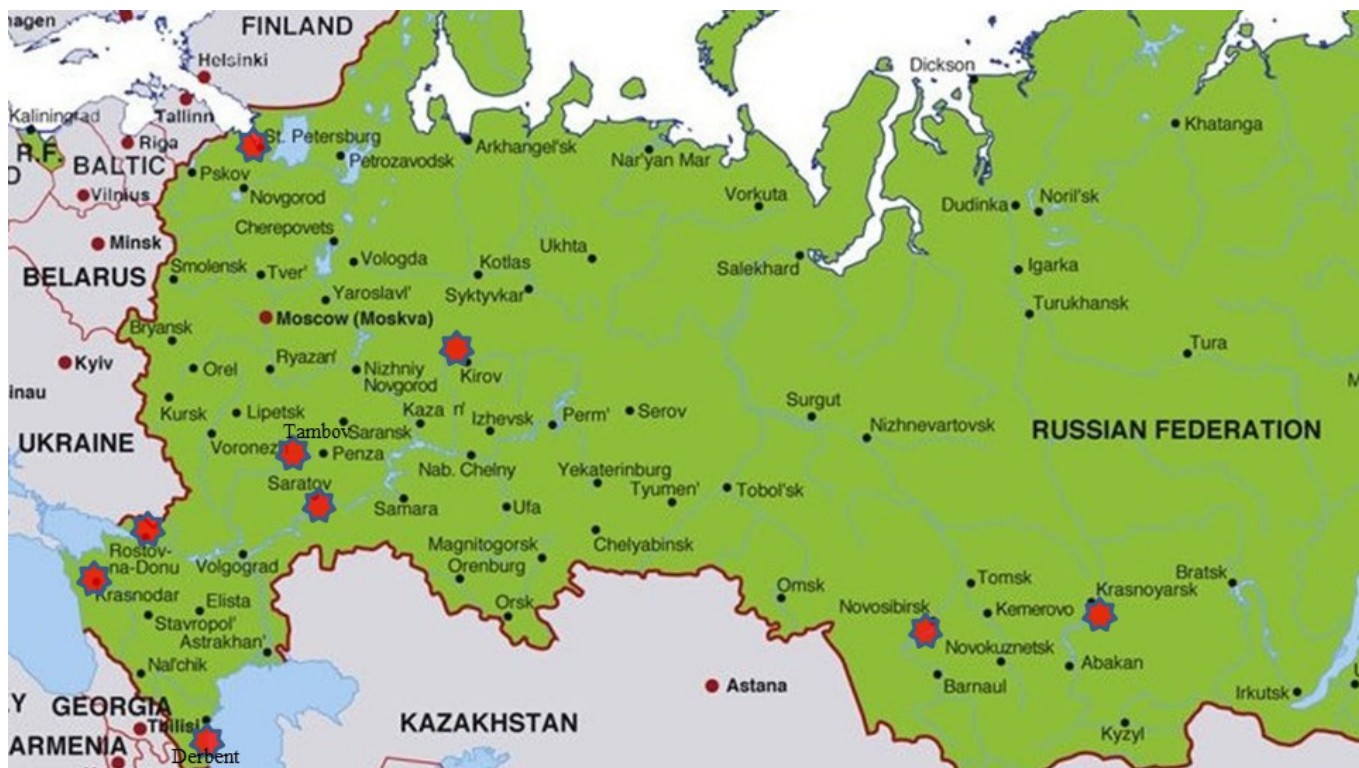

**Figure 1.** Collection sites of *Puccinia striiformis* in Russia.

**Table 1.** Sampling and isolation details for collections of *Puccinia striiformis* in regions of Russia.

| Region | Location | Year | Samples [1] | Isolates [2] |
|---|---|---|---|---|
| North Caucasus (NC) | Dagestan (D) | 2020 | 5 | 8 |
| | | 2021 | 19 | 18 |
| | | Total Da | 24 | 26 |
| | Krasnodar (Kr) | 2019 | 4 | 4 |
| | | 2020 | 5 | 8 |
| | | 2021 | 5 | 7 |
| | | Total Kr | 14 | 19 |
| | Rostov (Ro) | 2021 | 6 | 6 |
| | | Total NC | 44 | 51 |
| North-West (NW) | Saint Petersburg (SPb) | 2019 | 8 | 8 |
| | | 2020 | 15 | 23 |
| | | 2021 | 11 | 11 |
| | | Total NW | 34 | 42 |
| Low Volga (LV) | Saratov (Sa) | 2020 | 2 | 2 |
| | | 2021 | 2 | 2 |
| | | Total LV | 4 | 4 |
| Central Black Earth region (CBE) | Tambov (Ta) | 2020 | 2 | 2 |
| Volga-Vyatka (VV) | Kirov (Ki) | 2020 | 2 | 4 |

**Table 1.** *Cont.*

| Region | Location | Year | Samples [1] | Isolates [2] |
|---|---|---|---|---|
| West Siberia (WS) | Novosibirsk (No) | 2019 | 4 | 4 |
| | | 2020 | 2 | 4 |
| | | 2021 | 2 | 4 |
| | | Total No | 8 | 12 |
| | Krasnoyarsk (Kk) | 2021 | 1 | 4 |
| | | Total WS | 9 | 16 |
| | | Total Russia | 95 | 117 |

[1] Samples of stripe-rust-infected leaves (1 to 10) were collected from commercial fields and nurseries throughout wheat-growing areas; [2] single pustule isolates.

Mostly, one or two single uredinial isolates were obtained from each rust sample and tested for infection type. However, in the case of bulk-infected leaves from commercial fields, virulence pathotypes of at least three single pustule isolates were determined. Common white winter wheat cultivar Michigan Amber, susceptible to all *Pst* isolates at the seedling stage, was inoculated with urediniospores from samples to obtain initial isolates.

Standard procedures and conditions with some modifications were used for inoculating, growing plants before and post inoculation, and collecting urediniospores [16,33,34]. To recover urediniospores, a section of leaf (3–5 cm long) with uredinia from each sample was incubated in a Petri dish at 3–5 °C. The lower part of the leaf segments was covered with cotton wool with benzimidazole solution (0.004%). In 1 to 5 days, pieces of single lesions with fresh urediniospores were tied with plastic film onto 10–12-day-old susceptible wheat cultivar Michigan Amber. Inoculated plants were kept in a dark dew chamber at 10 °C for 24 h. Then, the film with infected leaf segments was removed and the pots transferred to a growth chamber (Environmental Test Chamber MLR-352H, Sanyo Electric Co., Ltd., Osaka, Japan) at 10 °C and 16:8 h L:D photoperiod. To prevent cross contamination, plants inoculated with different isolates were isolated with perforated cellophane bags. Approximately 14 to 21 days after inoculation, first sporulation was evident on inoculated leaves [34]. Urediniospores were collected 18–20 days after inoculation and once every 4 days until leaf desiccation. Cyclone spore collector was used for urediniospores harvesting. Fresh urediniospores, or those stored at 4 °C for less than 2 weeks, were used for testing on the host differentials.

*2.2. Virulence Characterization*

The virulence of each isolate was determined by testing on the set of 12 wheat *Yr* single-gene differentials in the Avocet spring wheat background and on eight supplemental wheat differentials (Table 2). Two cultivars, Jupateco S and Avocet S, were used as susceptible controls. Urediniospores of a single isolate was suspended in 5 mL Novec 7100 in a glass tube and connected to the airbrush spray gun. This suspension was sprayed onto three seedlings (2-leaf stage) of each differential line. The inoculated plants were incubated in a dark dew chamber at 10 °C for 24 h and transferred to a controlled-climate chamber, with the parameters described above. Seedling infection type was scored after 20 days on a five-point scale [35]: 0 to 2, host resistant (pathogen-avirulent) and 3 to 4, host susceptible (pathogen-virulent) [36].

**Table 2.** Virulence frequency of *Puccinia striiformis* collections of isolates originated from common wheat in Russian regions.

| *Yr* Genes | Line with the Corresponding *Yr* Genes | NC [1] | | | NW | LV | CBE | VV | WS | | Overall |
|---|---|---|---|---|---|---|---|---|---|---|---|
| | | D [2] | Kr | Ro | SPb | Sa | Ta | Ki | No | Kk | |
| *Yr1* | *Yr1*/6*Avocet S, Chinese 166 | 42 | 95 | 100 | 55 | 50 | 100 | 100 | 100 | 100 | **68** |
| *Yr2, Yr25, Yr+* [3] | Heines VII | 96 | 84 | 100 | 100 | 50 | 100 | 100 | 92 | 100 | **94** |
| *Yr3, Yr+* | Vilmorin 23 | 69 | 89 | 100 | 79 | 0 | 100 | 100 | 75 | 100 | **78** |
| *Yr4, Yr+* | Hybrid 46 | 88 | 95 | 0 | 62 | 50 | 0 | 100 | 83 | 100 | **73** |
| *Yr5* | *Yr5*/6*Avocet S | 0 | 0 | 0 | 0 | 0 | 0 | 0 | 0 | 0 | **0** |
| *Yr6* | *Yr6*/6*Avocet S | 92 | 100 | 100 | 100 | 100 | 100 | 100 | 100 | 100 | **98** |
| *Yr7* | *Yr7*/6*Avocet S | 100 | 95 | 33 | 95 | 50 | 50 | 100 | 50 | 100 | **86** |
| *Yr8* | *Yr8*/6*Avocet S | 100 | 95 | 100 | 98 | 100 | 100 | 0 | 50 | 0 | **88** |
| *Yr9* | *Yr9*/6*Avocet S | 85 | 100 | 100 | 100 | 50 | 100 | 100 | 100 | 100 | **95** |
| *Yr10* | *Yr10*/6*Avocet S | 0 | 0 | 0 | 0 | 0 | 0 | 0 | 0 | 0 | **0** |
| *Yr15* | *Yr15*/6*Avocet S | 0 | 0 | 0 | 0 | 0 | 0 | 0 | 0 | 0 | **0** |
| *Yr17* | *Yr17*/6*Avocet S | 0 | 0 | 0 | 12 | 25 | 0 | 0 | 0 | 0 | **5** |
| *Yr24* | *Yr24*/6*Avocet S | 0 | 0 | 0 | 0 | 0 | 0 | 0 | 0 | 0 | **0** |
| *YrSD, Yr25, Yr+* | Strubes Dickkopf (SD) | 46 | 26 | 50 | 74 | 100 | 0 | 100 | 100 | 100 | **62** |
| *Yr27* | *Yr27*/6*Avocet S | 100 | 63 | 0 | 67 | 100 | 100 | 100 | 83 | 100 | **75** |
| *Yr32, Yr25, Yr+* | Carstens V | 58 | 74 | 0 | 67 | 75 | 50 | 100 | 83 | 100 | **66** |
| *YrSu, Yr+* | Suwon 92/Omar (Su) | 92 | 100 | 67 | 95 | 100 | 100 | 100 | 100 | 100 | **95** |
| *YrSp* | *YrSP*/6*Avocet S | 50 | 58 | 100 | 29 | 0 | 0 | 100 | 67 | 100 | **48** |
| *YrND, Yr3* | Nord Desprez (ND) | 15 | 0 | 0 | 17 | 50 | 50 | 100 | 33 | 100 | **21** |
| *Yr2,Yr6,Yr25, Yr+* | Heines Peko (HP) | 92 | 100 | 100 | 88 | 50 | 50 | 100 | 100 | 100 | **91** |
| *Number of isolates* | | 26 | 19 | 6 | 42 | 4 | 2 | 2 | 12 | 4 | 117 |

[1] NC—North Caucasus, NW—North-West, LV—Low Volga, CBE—Central Black Earth region, VV —Volga-Vyatka, WS—West Siberia; [2] D—Dagestan, Kr—Krasnodar, Ro—Rostov, SPb—St. Petersburg, Sa—Saratov, Ta—Tambov, Ki—Kirov, No—Novosibirsk, and Kk—Krasnoyarsk; [3] *Yr+* corresponds to additional unidentified genes [33].

### 2.3. Data Analysis

*Pst* races, their frequencies and distribution, and frequencies of virulence to *Yr* genes or differential genotypes were analyzed. Original virulence data were clone-corrected so that isolates with identical virulence pathotypes at each regional population were represented by one unique *Pst* pathotype. Variability within and among the *Pst* populations was further analyzed for such clone-corrected data.

Descriptive parameters, such as virulence frequency and relative virulence complexity [37], were calculated for each clone-corrected regional population (virulence frequency was also estimated on the basis of isolate composition, i.e., considering pathotype abundances). Structure and relationships among the *Pst* populations were analyzed using the assignment-based approach applied to the simple mismatch dissimilarities between virulence pathotypes. The corresponding Kosman dispersion within ($KW$) and Kosman distance between populations ($KB$) were calculated [38–40]. Differentiation among populations was estimated with the permutation test (1000 random partitions) for differentiation statistics $dif_{KW}$ (Equation (1) in [34,41] based on the $KW$ dispersion (see also [40], Equation (13); and [42], p. 565).

Number of effectively different pathotypes within a population was estimated with the metric of functional trait dispersion [43] (Equations (5) and (6) for $M = KW$). Since the effective number depends on an actual number of pathotypes in a given population,

the normalized version of this indicator was considered (corrected Equation (5) in [44]; Equation (3) in [45]).

Singularity of each virulence pathotype within every regional collection of *Pst* was estimated on the basis of the simple mismatch dissimilarity between the pathotypes (Equations (1)–(3) in [46]). Then, the singularity of a population was calculated as the average singularity of all pathotypes that belong to that population (Equations (7) and (8) in [47]).

Unweighted pair–group method with arithmetic mean (UPGMA) was used for analyzing relationships among the *Pst* virulence pathotypes. UPGMA dendrograms of relationships among the *Pst* virulence pathotypes with regard to the simple mismatch dissimilarity were generated using the SAHN program of the NTSYSpc package, version 2.2 (Exeter Software, Setauket, NY, USA). Non-metric multi-dimensional scaling (NMDS) plots with regard to the Kosman distance (*KB*) between the regional *Pst* populations were derived using the Mdscale program of the NTSYSpc package, version 2.2 (Exeter Software, Setauket, NY, USA). Other above-mentioned calculations were performed with VIRULENCE ANALYSIS TOOL (VAT) software [47,48] and its extension.

## 3. Results

### 3.1. Virulence Characterization

In total, 117 *Pst* isolates obtained from common wheat in geographically dispersed regions of Russia were tested. The number of sampled isolates varied from two in Kirov and Tambov collections to 42 in St. Petersburg (Table 2). Among the 20 *Yr* genes, virulence to 16 of them was detected. All isolates were avirulent on *Yr5*, *Yr10*, *Yr15*, and *Yr24*, i.e., these genes were effective against the Russian *Pst* populations (Table 2). High virulence frequencies (≥80%) were observed on Avocet lines with *Yr6* and *Yr9* genes, as well as with cultivars Heines VII, Suwon 92/Omar, and Heines Peko. Most isolates were virulent in lines with *Yr27* (≥63%), cvs Carstens V (≥50%), and Vilmorin 23 (≥67%), with the exception of one or two isolates from Rostov and Saratov in each case. Virulence to *Yr17* was found only in *Pst* isolates from St. Petersburg and Saratov. Variation in virulence frequency among all *Pst* isolates was observed on other differential lines.

### 3.2. Pathotypes Composition

On the basis of the set of 20 differential lines and Avocet NILs, 79 *Pst* virulence pathotypes (phenotypes, races) were identified among 117 isolates (Tables 3 and 4). There were 5 phenotypes among 7 isolates detected in 2019, as well as 44 phenotypes among 62 isolates in 2020 and 37 phenotypes among 48 isolates in 2021. The number of pathotypes ranged from 1 (Kirov and Krasnoyarsk) to 36 (St. Petersburg). No single phenotype was detected across all locations. Predominant pathotypes P1, P2, and P3 were common in populations from European part of Russia. Pathotypes P1 and P2 were detected in three locations: Dagestan, Krasnodar, and St. Petersburg. Pathotype P3 was found in Dagestan and St. Petersburg. The common pathotype P4 was detected in Kirovsk and Krasnoyarsk, where *Pst* samples were collected only from commercial fields. Thirteen other pathotypes (P5 to P17) were collected as two and more isolates in one location. Other pathotypes (62) were detected as single isolate in one year only. High diversity of *Pst* pathotypes composition can be provided by the genetic diversity of wheat cultivars collected in various locations, such as breeding nurseries and experimental plots.

Descriptive parameters of the *Pst* isolates and virulence of predominant pathotypes are presented in Table 4. Frequency of predominant pathotypes varied from 7 to 100%. The average virulence complexity of pathotypes from North Caucasus, North-West, and Central European regions was significantly smaller than that from West Siberia (10.6, 11.1, 10.3, and 12.4, respectively). Virulence patterns of more distributed pathotypes are presented in Table 4.

**Table 3.** Number of isolates and number of pathotypes of *Puccinia striiformis* identified and their frequencies in various locations in 2019–2021.

| Region | Location | Year | Isolates | Pathotypes | Pathotype Composition and Frequency |
|---|---|---|---|---|---|
| NC | D | 2020 | 8 | 6 | **P1** [1] (38%), **P5** (12%), P18 (12%), P19 (12%), P20 (12%), P21 (12%), |
| | | 2021 | 18 | 14 | **P1** (17%), **P2** (5%), **P3** (5%), **P5** (5%), P6 (11%), P7 (11%), P22 (5%), P23 (5%), P24 (5%), P25 (5%), P26 (5%), P27 (5%), P28 (5%), P29 (5%) |
| | Kr | 2019 | 3 | 3 | **P1** (33%), P30 (33%), P31 (33%) |
| | | 2020 | 8 | 6 | **P2** (25%), **P8** (25%), P32 (12%), P33 (12%), P34 (12%), P35 (12%) |
| | | 2021 | 7 | 6 | **P2** (29%), **P1** (14%), **P8** (14%), P36 (14%), P37 (14%), P38 (14%) |
| | R | 2021 | 6 | 6 | P39 (17%), P40 (17%), P41 (17%), P42 (17%), P43 (17%), P44 (17%) |
| NW | SPb | 2019 | 8 | 7 | **P9** (25%), **P1** (12%), P45 (12%), P46 (12%), P47 (12%), P48 (12%), P49 (12%), P50 (12%) |
| | | 2020 | 23 | 22 | **P9** (9%), **P1** (4%), **P2** (4%), **P3** (4%), P10 (8%), P51 (4%), P52 (4%), P53 (4%), P54 (4%), P55 (4%), P56 (4%), P57 (4%), P58 (4%), P59 (4%), P60 (4%), P61 (4%), P62 (4%), P63 (4%), P66 (4%), P65 (4%), P66 (4%), P67 (4%), P68 (4%) |
| | | 2021 | 11 | 8 | **P9** (18%), P11 (18%), P12 (18%), P69 (9%), P70 (9%), P71 (9%), P72 (9%), P73 (9%) |
| LW | Sar | 2020 | 2 | 2 | P74 (50%), P75 (50%) |
| | | 2021 | 2 | 1 | P13 (100%) |
| CBE | Tam | 2020 | 2 | 2 | P76 (50%), P77 (50%) |
| VV | Kir | 2020 | 2 | 1 | **P4** (100%) |
| WS | Nov | 2019 | 3 | 2 | P14 (67%), **P15** (33%) |
| | | 2020 | 4 | 2 | **P15** (50%), P16 (50%) |
| | | 2021 | 4 | 3 | P17 (50%) P78 (25%), P79 (25%) |
| | Kr-k | 2020 | 4 | 1 | **P4** (100%) |

[1] Predominant phenotypes are indicated in bold.

**Table 4.** Pathotypes of *Puccinia striiformis* identified among isolates originating from common wheat in Russian regions.

| Parameters | NC | | | | NW | | C | | | WS | | | Overall, for Russia |
|---|---|---|---|---|---|---|---|---|---|---|---|---|---|
| | D | Kr | Ro | Total | SPb | Sa | Ta | Ki | Total | No | Kk | Total | |
| Number of isolates | 26 | 19 | 6 | 51 | 42 | 4 | 2 | 2 | 8 | 12 | 4 | 16 | **117** |
| Number of pathotypes | 18 | 11 | 6 | 34 | 36 | 3 | 2 | 1 | 6 | 7 | 1 | 8 | **79** |
| Predominant abundances [1], % | 23.1 | 21.1 | 16.7 | 17.6 | 7.1 | 50 | 50 | 100 | 25 | 25 | 100 | 25 | **9.4** |
| Average virulence complexity of pathotypes | 10.7 | 11.3 | 9.5 | 10.6 | 11.1 | 9.3 | 10 | 14 | 10.3 | 11.8 | 14 | 12.4 | **10.9** |
| Prevailing *Pst* pathotypes (P#: avirulence/virulence formula of *Yr* genes) and their abundances (%): | | | | | | | | | | | | | |
| P1. *Yr: 5,10,15,17, 24,ND/1,2,3, 4,6,7,8,9,25,27,32,Su,SP,HP* | 23 | 16 | 0 | 18 | 5 | 0 | 0 | 0 | 0 | 0 | 0 | 0 | 9 |
| P2. *Yr: 5,10,15,17,24,25,32,Sp, ND/1,2, 3,4,6,7,8,9,27,Su,HP* | 4 | 21 | 0 | 10 | 2 | 0 | 0 | 0 | 0 | 0 | 0 | 0 | 5 |
| P3. *Yr: 1,5,10,15,17,24,32, SP, ND/2,3,4,6,7,8,9,25,27,Su,HP* | 4 | 0 | 0 | 3 | 2 | 0 | 0 | 0 | 0 | 0 | 0 | 0 | 2 |
| P4. *Yr: 5,8,10,15,17,24/1,2,3,4, 6,7,9,25,27,32,Su,Sp, ND,HP* | 0 | 0 | 0 | 0 | 0 | 0 | 0 | 0 | 0 | 0 | 100 | 0 | 4 |

**Table 4.** *Cont.*

| Parameters | NC | | | | NW | | C | | | WS | | | Overall, for Russia |
|---|---|---|---|---|---|---|---|---|---|---|---|---|---|
| | D | Kr | Ro | Total | SPb | Sa | Ta | Ki | Total | No | Kk | Total | |
| P5. *Yr: 5,9,10,15,17,24,25,ND/1,2,3,4,6,7,8,27,32,Su,SP,HP* | 4 | 0 | 0 | 4 | 0 | 0 | 0 | 0 | 0 | 0 | 0 | 0 | 2 |
| P6. *Yr: 1,3,5,10,15,17,24,25,32,Sp,ND/2,4,6,7,8,9,27,Su, HP* | 8 | 0 | 0 | 4 | 0 | 0 | 0 | 0 | 0 | 0 | 0 | 0 | 2 |
| P7. *Yr: 1,3,5,10,15,17,24,32,SP,ND/2,4,6,7,8,9,25,27, Su,HP* | 8 | 0 | 0 | 4 | 0 | 0 | 0 | 0 | 0 | 0 | 0 | 0 | 2 |
| P8. *Yr: 5,10,15,17,24,25,SP,ND/1,2,3,4,6,7,8,9,27, 32,Su, HP* | 0 | 16 | 0 | 6 | 0 | 0 | 0 | 0 | 0 | 0 | 0 | 0 | 3 |
| P9. *Yr: 5,10,15,17,24,SP,ND/1,2,3,4,6,7,8,9,25,27, 32,Su,HP* | 0 | 0 | 0 | 0 | 7 | 0 | 0 | 0 | 0 | 0 | 0 | 0 | 3 |
| P10. *Yr: 1,4,5,10,15,17,24,SP,ND/2,3,6,7,8,9,25,27,32,Su,HP* | 0 | 0 | 0 | 0 | 5 | 0 | 0 | 0 | 0 | 0 | 0 | 0 | 2 |
| P11. *Yr: 1,5,10,15,17,24,SP,ND/2,3,4,6,7,8,9,25,27, 32,Su,HP* | 0 | 0 | 0 | 0 | 5 | 0 | 0 | 0 | 0 | 0 | 0 | 0 | 2 |
| P12. *Yr: 4,5,10,15,17,24,SP/1,2,3,6,7,8,9,25,27,32,Su,ND,HP* | 0 | 0 | 0 | 0 | 5 | 0 | 0 | 0 | 0 | 0 | 0 | 0 | 2 |
| P13. *Yr: 2,3,5,7,10,15,17,24,SP,ND/1,4,6,8,9,25,27,32,Su,HP* | 0 | 0 | 0 | 0 | 0 | 50 | 0 | 0 | 0 | 0 | 0 | 0 | 2 |
| P14. *Yr: 3,5,8,10,15,17,24,ND/1,2,4,6,7,9,25,27, 32,Su, SP,HP* | 0 | 0 | 0 | 0 | 0 | 0 | 0 | 0 | 0 | 25 | 0 | 0 | 3 |
| P15. *Yr: 5,8,10,15,17,24,ND1,2,3,4,6,7,9,25,27,32,Su,SP,HP* | 0 | 0 | 0 | 0 | 0 | 0 | 0 | 0 | 0 | 17 | 0 | 0 | 2 |
| P16. *Yr: 5,7,10,15,17,24/1,2,3,4,6,8,9,25,27,32,Su,SP,ND,HP* | 0 | 0 | 0 | 0 | 0 | 0 | 0 | 0 | 0 | 17 | 0 | 0 | 2 |
| P17. *Yr: 5,7,10,15,17,24,27,SP,ND/1,2,3,4,6,8,9,25,32,Su,HP* | 0 | 0 | 0 | 0 | 0 | 0 | 0 | 0 | 0 | 17 | 0 | 0 | 2 |

[1] Frequency of predominant pathotypes (%).

### 3.3. Variability within and among Regional Collections of Pst Pathotypes

Variability within the regional Pst populations differed considerably. Population from the Central region was the most variable, with dispersion KW = 0.36 and the normalized effective number of different pathotypes (Table 5). The least variable population was from Rostov (KW = 0.12).

**Table 5.** Variability within Puccinia striiformis populations sampled from common wheat in Russian region.

| *Pst* Collection | Pathotypes | KW | [1]D(T,KW) | [1]nD(T,KW) |
|---|---|---|---|---|
| Krasnodar | 11 | 0.21 | 2.99 | 0.12 |
| Rostov | 6 | 0.12 | 1.56 | 0.11 |
| Dagestan | 18 | 0.28 | 5.49 | 0.26 |
| North-West | 36 | 0.31 | 11.37 | 0.3 |
| West Siberia | 8 | 0.27 | 2.77 | 0.25 |
| Central European regions | 6 | 0.36 | 2.41 | 0.35 |

The relatively high variability found within Dagestan, Central, North-West, and West Siberia regions (Table 5) provided evidence of possible subdivision within each region.

Indeed, Pst pathotypes in Dagestan, Central, North-West, and West Siberia regions were separated into three, two, five, and two subgroups, respectively, on the basis of clustering of pathotypes within each population by UPGMA dendrograms using simple mismatch dissimilarity (data not shown). The number of pathotypes in each subgroup is shown in Table 6.

**Table 6.** Virulence pathotypes in subgroups within regional Puccinia striiformis populations.

| Regional Population | Abbreviation | Number of Virulence Pathotypes in Five Subgroups | | | | | Total |
|---|---|---|---|---|---|---|---|
| | | 1 | 2 | 3 | 4 | 5 | |
| Dagestan | D | 11 | 4 | 3 | | | 18 |
| North-West | NW | 4 | 11 | 12 | 7 | 2 | 36 |
| West Siberia | WS | 6 | 2 | | | | 8 |
| Central | C | 2 | 3 | | | | 6 |

The values given under each subgroup (1–5) are the numbers of virulence pathotypes in each region.

Relationships between the regional Pst populations and subgroups based on the KB distance with regard to the simple mismatch dissimilarity between them can be uncovered from the NMDS plot in Figure 2. Two subgroups from Dagestan (D_1 and D_2; altogether 15 of 18 pathotypes) were closely related to three subgroups from the North-West (NW_2, NW_3 and NW_4; altogether 30 of 36 pathotypes) that can explain insignificance of differentiation among populations from Dagestan and North-West at $p = 0.05$ (1–0.9 = 0.06 > 0.05; Table 7). Most virulence pathotypes from West Siberia (6 of 8 in subgroup WS_1) were closely related to pathotypes from the most European and North Caucasian Pst subgroups (Figure 2). Several small groups of pathotypes (up to 3 in each of group C_1, D_3, NW_5 and WS_2) were considerably different from all other Pst subgroups (Figure 2).

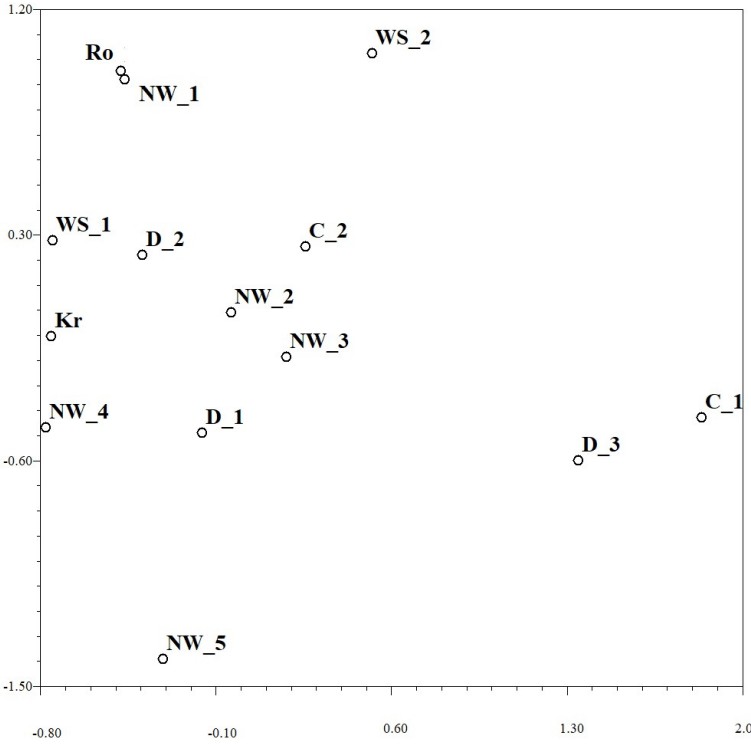

**Figure 2.** Non-metric multi-dimensional scaling plot based on the *KB* distance with regard to the simple mismatch dissimilarity between *Puccinia striiformis* collections of pathotypes from different regions. Abbreviations of regions and subgroups: Kr—Krasnodar, Ro—Rostov, D—Dagestan, NW—North-West, WS—West Siberia, and C—Central European regions; NW_1 means the first subgroup in NW region.

**Table 7.** Pairwise distances (*KB*; below diagonal) and significance of differentiation (*KW*; above diagonal) between Russian regional *Puccinia striiformis* populations.

|  | **Kr** | **Ro** | **D** | **NW** | **WS** | **C** |
|---|---|---|---|---|---|---|
| **Kr [a]** | - | 0.99 | 0.99 | 0.98 | 0.97 | 0.99 |
| **Ro [b]** | 0.22 | - | 0.99 | 0.99 | 0.99 | 0.99 |
| **D** | 0.17 | 0.29 | - | 0.94 | 0.99 | 0.97 |
| **NW** | 0.16 | 0.24 | 0.14 | - | 0.38 | 0.91 |
| **WS** | 0.18 | 0.25 | 0.23 | 0.2 | - | 0.89 |
| **C [b]** | 0.24 | 0.32 | 0.19 | 0.2 | 0.22 | - |

[a] Kr—Krasnodar, Ro—Rostov, D—Dagestan, NW—North-West, WS—West Siberia, and C—Central European regions; [b] Number of isolates was relatively small (8 and 6, respectively).

In each region, singular *Pst* virulence pathotypes were characterized by relatively low virulence complexity that did not exceed the average for that region. These did not include any uncommon virulence pathotypes, so these singular pathotypes were weak with no signs of being particularly harmful or aggressiveness for common wheat cultivars.

## 4. Discussion

In the present study, we analyzed virulence of stripe rust samples from common wheat in the major wheat production regions of Russia in 2019–2021. These regions were distant from each other, extending approximately 2300 and 3600 km from Saint Petersburg (North-West) to Derbent (Dagestan) and Krasnoyarsk (West Siberia), respectively, and more than 3800 km between Krasnodar (North Caucasus) and Krasnoyarsk. Winter wheat dominates in North Caucasus, whereas spring wheat is grown in West Siberia and Low Volga. In the North-West and Central European regions, both winter and spring wheat are cultivated [49]. Except for North Caucasus, wheat stripe rust has, until recently, occurred rarely only in these regions.

We found that virulence variation was relatively high within the Russian populations. Seventy-nine pathotypes among 117 isolates were identified using the 12 Avocet *Yr* gene lines and eight supplemental wheat differentials. Among these, 78% were detected only as a single isolate. Mostly closely related races differed in virulence due to a single resistance gene in the host. Common pathotypes were detected in widely separated North Caucasus and North West; and West Siberia and Volga-Vyatka regions.

The populations of *Pst* are characterized by high diversity around the world. Sharma-Poudyal et al. [27] studied 235 isolates from Algeria, Australia, Canada, Chile, China, Hungary, Kenya, Nepal, Pakistan, Russia, Spain, Turkey, and Uzbekistan and identified 129 virulence patterns in 20 single-gene lines and 169 virulence patterns in the 20 US differentials. In 2011 and 2013, *Pst* collections from Saskatchewan (Canada) and southern Alberta (USA) were analyzed by Brar et al. [17] Virulence analyses differentiated the 59 isolates into 33 races, of which 26 were represented by single isolates. In 2017–2019, 62 *Pst* isolates were studied in Israel. There were 32 virulence phenotypes detected, with 30 of them being unique phenotypes [50]. A similar result was described for many other world *Pst* collections [16,51]. Diversity of Russian North-Caucasian *Pst* population from Krasnodar, Stavropol, and Rostov was evaluated in 2013–2018 by Volkova et al. [32]. One hundred and eighty-two virulence phenotypes were identified in 186 *Pst* isolates. The isolates collected in 2014, 2015, and 2018 were all unique pathotypes. In the present study conducted in 2019–2021, the diversity of the North Caucasian population was substantially lower (34 phenotypes among 51 isolates) (Table 4).

High variability was detected within Dagestan, Central, North-West, and West Siberia *Pst* populations that could be subdivided within each population into several subgroups. Mostly pathotypes from these subgroups were avirulent to ineffective genes. For example, North-Western pathotypes from contrasting *Pst* Subgroup 5 differed from other North-West

subgroups by avirulence to *Yr7*; Dagestan (Subgroup 3) and West Siberian (Subgroup 2) pathotypes by avirulence to *Yr4*; and Central European pathotypes (Subgroup 1) by avirulence to *Yr3*, *Yr4*, and *Yr9*. These subgroups were distinct from all other regional *Pst* subgroups. Most other virulence groups of pathotypes from North Caucasus, North-West, and Central and West Siberia regions were closely related.

Breeding wheat for yellow rust resistance has been a priority over an extended period in the North Caucasus region where the disease has high importance. In the North Caucasus, as in other Russian regions, cultivars with seedling resistance gene *Yr9* and partial resistance gene (or slow-rusting gene) *Yr18* are widely grown. In the North Caucasus, as in other Russian regions, cultivars with seedling resistance gene *Yr9* and partial resistance gene (or slow-rusting gene) *Yr18* are widely grown. Currently, wheat cultivars with *Yr9* gene are mostly susceptible to yellow rust worldwide. The resistance breakdown of *Yr9* has happened since the late 1980s due to widespread emergence of virulent races worldwide; this has resulted in major epidemics of stripe rust that have challenged world wheat production [3,4]. In Russia, high virulence frequency to *Yr9* was found in all the *Pst* populations we studied.

The slow-rusting gene *Yr18/Lr34/Sr57* confers partial durable resistance to multiple fungal pathogens, including yellow rust. Other examples of such genes included *Yr29/Lr46/Sr58/Pm39*, *Yr30/Lr27/Sr2*, and *Yr46/Lr67/Sr55/Pm46*. This resistance type is characterized by slow rusting (an extended period of latent infection and small lesion size) [52–55]. Under field conditions, Russian wheat cultivars with *Yr18* gene were either moderately or highly susceptible to *Pst* at the adult plant stage, depending on genetic background of a given cultivar (e.g., Adel, Bagira, Bezostaya 1, Bezostaya 100, Don 174, Kuma, and Rostovchanka 3). At the same time, the transmission of infection from some of these cultivars on susceptible wheat seedlings was not always successful, which indicates a weaker virulence and aggressiveness of *Pst* isolates that affect genotypes with *Yr18*.

Wheat cultivars with *Yr17* gene are grown in North Caucasus and Central European regions and most of them have a resistant reaction to *Pst* [56]. The planting area of cultivars with *Yr17* gene was smaller than for cultivars with *Yr9* and *Yr18* genes [57]. *Yr17* resistance gene has been incorporated into wheat cultivars in northern Europe since the mid-1970s. Wheat cultivars with *Yr17* resistance were first grown in the UK, Denmark, France, and Germany in 1980–1990 [58]. Virulence to *Yr17* was detected after 1995, following intensive use of this single resistance gene in widely grown cultivars [59]. Currently *Yr17*-virulence is common in northwestern European *Pst* populations. The Russian *Pst* isolates studied were mostly avirulent to *Yr17*, except several isolates of two pathotypes, one each from North-West and Low Volga regions, which perhaps arrived by wind dispersal from western Europe.

Virulences to *Yr5*, *Yr10*, *Yr15*, and *Yr24* genes were not known in Russia in 2019–2021. Sharma-Poudyal et al. [27] did not reveal isolates virulent to these genes in studies of two Russian *Pst* collections in 2006. Resistance genes *Yr5* and *Yr15* are still effective against the predominant *Pst* races in many countries [21,27,60–63]. Virulence to *Yr10* and *Yr24* occurs in some regional populations around the globe, but with low frequency [26,27]. According a GRRC report on yellow rust genotyping and race analysis in 2021 [26], virulences to *Yr10* and *Yr24* are more typical for races considered as *Triticale*-aggressive (*PstS4* and *PstS13*), which are found in Europe, and among isolates belonging to the *PstS2* genetic group. We compared virulence of Russian *Pst* pathotypes from this study with virulence of *PstS* race groups described in the GRRC study [26]. Differential set with genes *Yr1*, *Yr2*, *Yr3*, *Yr4*, *Yr5*, *Yr6*, *Yr7*, *Yr8*, *Yr9*, *Yr10*, *Yr15*, *Yr17*, *Yr24*, *Yr25*, *Yr27*, and *Yr32* and cv. Spalding Prolific were used in both studies. Only one pathotype from North-West was common for virulence against the *PstS* 2v27 race.

There are several reasons why high variation in virulence occurs in Russian *Pst* populations. The first is long-distance dispersal of the *Pst* pathogen by wind [63]. Frequent outbreaks of the disease in the bordering countries, such as Azerbaijan, Georgia, Kazakhstan [64,65], and Latvia [66], may have had significant effects on the Russian *Pst*

populations. Secondly, overwintering of *Pst* on wild cereals and grasses, providing a green bridge (for example in North Caucasus), could increase *Pst* diversity [67]. In addition, the sexual stage of *Pst* reproducing in *Berberis* spp. results in recombination, increasing genetic variation and development of new virulence pathotypes. This alternate host of the yellow rust pathogen occurs widely in wheat production regions of Russia. Sinha and Chen [11] demonstrated that potential infection risks of the sexual reproduction of wheat stripe rust pathogen in *Berberis* in Russia varies among regions ranging from low to high risk.

## 5. Conclusions

Large-scale analysis of *Pst* populations from common wheat collected in geographically distant Russian regions was performed for the first time in this study. Virulence variation was found to be relatively high within the Russian populations. Most *Pst* pathotypes sampled in European and Asian parts of Russia were closely related, except for several small subgroups of pathotypes from West Siberia, Dagestan, North-West, and Central European regions. All *Pst* isolates were avirulent in lines with *Yr5, Yr10, Yr15,* and *Yr24* genes. Russian *Pst* isolates were mostly avirulent to *Yr17*, except for several isolates of two pathotypes, one each from North-West and Low Volga regions. These effective *Yr* genes can be recommended for yellow rust resistance breeding in Russia. Identical pathotypes were determined in geographically distant Russian regions, which could be due to the long-distance dispersal capacity of the yellow rust pathogen.

**Author Contributions:** Conceptualization and data analysis design, E.G. and E.K.; methodology and performing experiments, E.G. and E.S.; data analysis, E.K.; interpretation of results, E.G., E.S., and E.K.; data acquisition and curation, E.G.; drafting the manuscript, E.G. and E.K. All authors have read and agreed to the published version of the manuscript.

**Funding:** This study was funded by the Russian Science Foundation (project number 19-76-30005).

**Institutional Review Board Statement:** Not applicable.

**Informed Consent Statement:** Informed consent was obtained from all subjects involved in the study.

**Data Availability Statement:** All data are provided in the manuscript.

**Acknowledgments:** We thank all colleagues from Krasnodar, Dagestan, Rostov, Novosibirsk, Tambov, Saratov, Kirovsk, Krasnoyarsk and Saint Petersburg for their excellent assistance in collecting and sending samples of yellow rust uredinia.

**Conflicts of Interest:** The authors declare no conflict of interest.

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
