# Peer review of "Virulence Diversity of Puccinia striiformis f. sp. Tritici in Common Wheat in Russian Regions in 2019–2021"

_agriculture, doi:10.3390/agriculture12111957_

Round 1

Reviewer 1 Report

This study identified 79 phenotypes among 117 isolates using the 12 Avocet Yr gene lines and eight supplemental wheat differentials. All Pst isolates were avirulent on lines with Yr5, Yr10, Yr15, and Yr24 genes. These Pst isolates were mostly avirulent to Yr17 except for several isolates of two phenotypes, one each from the North-West and Low Volga regions. These effective Yr genes can be recommended for yellow rust resistance breeding in Russia. Comments are listed in the following:

l  More details should be added in the Materials and Methods Section. For example, Eq. 1 to 5 should be detailed, not only cited references.

l  Abbreviations should be defined when they first appear, e.g., KW in the 160th line, UPGMA in the 170th line, et al.  Abbreviations should also be the same. The abbreviation of Dagestan in Table 1 and Table 2 is different.

l  The three-line table is better for an article paper.

Author Response

Dear reviewer,

Thank you for your positive assessment of our work.

Answers to comments.

 More details should be added in the Materials and Methods Section. For example, Eq. 1 to 5 should be detailed, not only cited references.

<< We disagree with the reviewer. There is a need to mention all employed approaches and provide more detail explanations some key issues, but it is impossible and would be very unusual to explain everything in detail including data analysis. Our study is not about methods. Even ‘a brief information’ for each metric and approach used would increase considerably (30% at least) the manuscript size and shift the focus from the main objectives of this research.

 Abbreviations should be defined when they first appear, e.g., KW in the 160th line, UPGMA in the 170th line, et al. 

<< We agree. The corresponding modifications were made in the revised version.

Abbreviations should also be the same. The abbreviation of Dagestan in Table 1 and Table 2 is different.

<< Corrected

The three-line table is better for an article paper.

All tables are prepared in accordance with MDPI instructions.

With kind regards,

Elena Gultyaeva

Reviewer 2 Report

Dear authors,

Stripe rust is one of the most important diseases of wheat. The analysis of virulence variation and race composition of Pst isolates is of great importance. The manuscript “Virulence Diversity of Puccinia Striiformis f. sp. Tritici in Common Wheat in Russian Regions in 2019-2021” describes assess virulence variability and race composition of Pst isolates infecting common wheat in distant Russian locations, and  compare the pathogen structure in regional populations.

I also found some problems in the manuscript, such as: there are some errors in the format of the manuscript.

I suggest this manuscript can be accepted only after minor revisions throughout the manuscript.

Detailed comments:

1. P1.L11, 12. Pst should be italicized. Please check the full text for the same issue.

2. P1.L16. Yr should be italicized. Please check the full text for the same issue.

3. P1.L41-43. “Yellow rust symptoms appears as a……glumes and awns on susceptible plants.” These are the fundamentals and can be removed.

4. P2. L56-57. Here it is suggested to add the literature application of Zhou et al. 2022.

Xinli Zhou*, Taohong Fang, Kexin Li, Kebing Huang, Chunhua Ma, Min Zhang, Xin Li, Suizhuang Yang, Runsheng Ren,* and Pingping Zhang*. Yield losses associated with different levels of stripe rust resistance of commercial wheat cultivars in China. Phytopathology. 2022,DOI: 10.1094/PHYTO-07-21-0286-R.

5. P5. L157. KW and KB should be spelled in full when they appear first.

6. P8.L260. “P. strii-formis” should be italicized.

7. P10.L308. “Lr18” should be Yr18.

8. P10.L308-310. Please list the names of varieties containing Yr18.

9. P10.L339. “The first is long-distance dispersal of Pst pathogen by wind [62, 63].” This sentence is incomplete. Please revise it.

10. P11.L359. “……which could be cold be due……”Please check for corrections.

11. Check the format of all references and organize them according to journal requirements.

Author Response

Dear reviewer,

We accepted all editorial-type changes and added data to table.

A revised version of our manuscript was prepared accordingly.

Thank you very much for the comments and suggestions to improve the manuscript.

With kind regards,

Elena Gultyaeva

Reviewer 3 Report

The structure of the study and the presentation of the results are appropriate. But although the studies covered a vast territory, only over 100 isolates were isolated and studied in 3 years. Many of the genotypes used in the study had long-used genes. Genotypes with the newest genes were not tested, this does not allow us to get a picture of the virulence of the populations studied for these genes. Such results are suitable for publication only at the local level.

Author Response

We disagree with the comments of Reviewer3 suggesting that the results presented are only suitable for publication at the local level. Yellow rust is a large problem for wheat on a global scale. In this regard, it is widely studied all over the world. Comparison of results around the world is important to discover its origin and migration to help in control the disease. Therefore, there are many articles described the Pst  study in large territories (for example, Chen et al. Virulence characterization of Puccinia striiformis f.sp. tritici collections from six countries in 2013 to 2020. Canadian Journal of Plant Pathology. 2021) and local areas (Alam, M.A.; Li, H.; Hossain, A.; Li, M. Genetic diversity of wheat stripe rust fungus Puccinia striiformis f. sp. tritici in Yunnan, China. Plants 2021) and they have been published in international journals.

The methodologies of yellow rust populations study are based on the experiences of virulence phenotyping at the Global Rust Reference Center (see https://agro.au.dk and Hovmøller M.S. et al. 2017 Race typing of Puccinia striiformis on wheat. 2017, DOI 10.1007/978-1-4939-7249-4_3). Standard and extended sets of wheat differential lines with genes  Yr1, Yr2, Yr3, Yr4, Yr5, Yr6, Yr7, Yr8, Yr9, Yr10, Yr15, Yr17, Yr24, Yr25, Yr27, Yr32, and the resistance specificity of Spalding Prolific (Sp), Avocet S (AvS) were used for race typing of P. striiformis isolates. These are exactly the genotypes that the reviewer dismisses as, "Many of the genotypes used in the study had long-used genes."

However, there is no additional generally accepted set of genes, so the reviewers comment appears ill-informed. Commercial local cultivars and other lines of special interest ("Genotypes with the newest genes") also may be included in the set. But additional sets vary significantly between studies. The use of new cultivars in the analysis is important where these samples are already cultivated or are planned to be used in breeding. Because these cultivars are not used in breeding in Russia, we did not consider it necessary to include them in the research.

The other comment was, "But although the studies covered a vast territory, only over 100 isolates were isolated and studied in 3 years". Limitation of work with yellow rust is the rapid death of urediniospores, in comparison with other wheat rust species (leaf and stem rust). In all world studies, the number of Pst isolates is not as high as P. triticina or P. graminis. So the scale of this work is within expectations and is not out of step with other studies.

Some examples:

Wan et al. Virulence characterization of wheat stripe rust fungus Puccinia striiformis f. sp. tritici in Ethiopia and evaluation of Ethiopian wheat germplasm for resistance to races of the pathogen from Ethiopia and the United States . Plant Disease,  2017,  101:73-80.

In total, 97 isolates were recovered from stripe rust samples collected in Ethiopia in 2013 and 2014.

 Brar G. S. and Kutcher H. R. Race characterization of Puccinia striiformis f. sp. tritici, the cause of wheat stripe rust, in Saskatchewan and Southern Alberta, Canada and virulence comparison with races from the United States. 2016. Plant Dis.

Fifty-nine isolates of P. striiformis f. sp. tritici, the majority of which were collected between 2011 and 2013 from Saskatchewan and southern Alberta, were analyzed for virulence frequency and diversity and compared with isolates characterized in the Pacific Northwest and Great Plains regions of the United States. In all, 31 wheat differentials, including 20 near-isogenic lines, and 10 supplemental wheat lines including some Canadian wheat variety and 1 triticale variety. The commercial wheat were added without any prior knowledge of  resistance but were included because they were most widely cultivated in 2005 to 2013.

Amil et al. Pathotype diversification in the invasive PstS2 clonal lineage of Puccinia striiformis f. sp. tritici causing yellow rust on durum and bread wheat in Lebanon and Syria in 2010–2011 //Plant Pathology, 2020. 69(4)

54 isolates were pathotyped  in Lebanon and Syria in 2010–2011. The 48 successfully multiplied Syrian isolates were pathotyped at the ICARDA-Tal Hadya station during the summer and autumn of 2011, and the six multiplied Lebanese isolates were pathotyped at INRA Versailles in high-confinement controlled-climate rooms in May 2014.The virulence profiles of these isolates were determined with a robust set of 43 differential lines selected from the collection of standard differential lines used worldwide. Most of the Yr genes were present in more than one tester genotype and some tester genotypes harbored several resistance genes.

Chen X. et al. (2021): Virulence characterization of Puccinia striiformis f. sp. tritici collections from six countries in 2013 to 2020, Canadian Journal of Plant Pathology

A total of 491 Pst isolates were obtained from six countries, including 16 from Canada (one in 2013, two in 2016 and 13 in 2017), 139 from China in 2016, 45 from Ecuador (33 in 2015 and 12 in 2016), two from Egypt in 2018, 167 from Italy (25 in 2014, 108 in 2016, 22 in 2017, three in 2018 and nine in 2020) and 122 from Mexico (13 in 2015 and 109 in 2016). A total of 138 Pst races, including 120 races that were not previously reported, were identified from stripe rust collections made from Canada, China, Ecuador, Egypt, Italy and Mexico in 2013–2020 using a set of 18 Yr single-gene differentials.

Kosman et al. (2022). Virulence Survey of Puccinia striiformis in Israel revealed considerable changes in the pathogen population during the period 2001 – 2019. Plant Disease.

A total of 353 urediniospore isolates of Puccinia striiformis f. sp. tritici (Pst) collected in Israel during 2001-2019 were analyzed. Pst pathogenicity was studied with a set of 20 differentials (17 Avocet and 3 other lines).

Thus, the number of isolates analyzed in our study and differential set used does not differ substantively from those used in other countries. Consequently, we firmly be believe our results are internationally relevant and consistent in methodology, and reject any suggestion that they are only of regional significance, and find such a suggestion somewhat inexplicable.